# Identifying social science engagement within agroecology: Classifying transdisciplinary literature with a semi-automated textual classification method

**Natalia Pinzón**[1‡]*, **Ryan E. Galt**[1,2‡], **Marcela Beatriz Baukloh Coronil**[3]

**1** Geography Graduate Group, College of Agriculture and Environmental Science, University of California, Davis, Davis, California, United States of America, **2** Agricultural Sustainability Institute, University of California, Davis, Davis, California, United States of America, **3** Universidad Nacional del Este, Hernandarias, Paraguay

‡ NP and REG are joint senior authors on this work.
* npinzon@ucdavis.edu

**Data Availability Statement:** All methodology files (Python Script) and results (Agroecology

## Abstract

Interdisciplinary and transdisciplinary fields of inquiry and action have been important academic frontiers in recent years. The field of agroecology is a prime example of transdisciplinarity. With roots in the biophysical sciences, social sciences, and peasant movements, publications in agroecology have been growing rapidly in recent decades. Here we explain a method—the script-expert adaptive classification (SEAC) method—that allows us to examine the engagements between agroecology and the social sciences by identifying publications within the agroecological literature that engage with social science at various levels. Using the term "agroecology" and its iterations, we gathered a corpus of agroecology literature up to and including 2019 with 12,398 unique publications from five publication databases—Scopus, Web of Science, Agricola, CAB Direct, and EconLit. Using the SEAC method we then classified each publication as engaged, partially engaged, and not engaged with social sciences and separated this Agroecology Corpus 2019 into three corpora: agroecology engaged with social sciences (with 3,125 publications), agroecology not engaged with social sciences (with 7,039 publications), and agroecology with uncertain engagement with social science (with 2,234 publications) or unclassifiable. This article explains the SEAC method in detail so other transdisciplinary scholars can replicate and/or adapt it for similar purposes. We also assess the SEAC method's value in identifying social science publications relative to the classification systems of the major multidisciplinary bibliographic databases, Scopus, and Web of Science. We conclude by discussing the strengths and weaknesses of the SEAC method and by pointing to further questions about agroecology and the social sciences to be asked of the corpora.

Database) will be available from the University of California eScholarship database (https://escholarship.org/uc/item/5hb78383).

**Funding:** The author(s) received no specific funding for this work.

**Competing interests:** The authors have declared that no competing interests exist.

## Introduction: Transdisciplinarity and classification challenges

The emergence and maturation of interdisciplinary and transdisciplinary fields of inquiry and action have been a hallmark of much academic work in recent years. In particular, a variety of engagements between the social sciences and biophysical sciences have blossomed in recent decades. Some of these have involved transdisciplinarity, or "the application of theories, concepts, or methods across disciplines with the intent of developing an overarching synthesis" [1]. The most transdisciplinary of these engagements have given rise to new fields like agroecology [2], political ecology [3], and sustainability science [4, 5], to name a few. Zimmerer terms these kinds of engagements "the environmental borderlands," referring to the "ideas, concepts, methodologies, and topical issues that bridge or interconnect human and biogeophysical systems and GIScience" [6]. Although Zimmerer is assessing these environmental borderlands within the discipline of geography (hence the focus on GIScience), the environmental borderlands concept is widely applicable to a number of transdisciplinary fields, including agroecology.

We began this project to examine in detail the engagement between agroecology and the social sciences. We define agroecology broadly, following Francis et al., "as the study of the whole food system, embracing both natural and social sciences, and emphasizing systems thinking and ecological principles" [7]. Through publication network analysis in a recent review, Mason et al. [8] find that the agroecology literature is citing from interdisciplinary sources and utilizing a "transdisciplinary base of knowledge." We draw on the definition of the social sciences by Bennett et al.: "Both classic and applied social sciences are used to study a diverse set of social phenomena, social processes, or individual attributes. The classic social science disciplines include sociology, anthropology, political science, geography, economics, history, and psychology. Applied social science disciplines include education, communication studies, development studies and law." [9]

We acknowledge that agroecology and the social sciences cannot easily be parsed apart, since the critical social sciences were an important part of agroecology's origins [2, 10–14] and have continued to make substantial contributions as the field has developed [7, 8, 15, 16]. Tomich et al. note, "methods and issues from social science (including economics) are as important to agroecology as are the more conventional agroecological approaches that are based on agricultural, ecological, and environmental sciences" [16]. Yet, research in agroecology ranges substantially from studying the levels of farm plots, fields, farming systems, and food systems [7, 17]. At the plot and field level, agroecology researchers sometimes draw system boundaries that preclude analysis of social phenomena. A review of the social and economic impacts of agroecology notes that "there is a lack of data concerning the socio-economic impact of agroecology" [18]. Thus, we could not treat all publications in agroecology as inherently engaged with the social sciences, since many are clearly not.

We realized that in order to detail the engagement between agroecology and the social sciences, we first needed to answer the research question that drives this paper: what method would allow for a more accurate classification of social science engagement for publications from transdisciplinary fields like agroecology? We first turned to the leading multidisciplinary bibliographic databases—Scopus and Web of Science (WoS)—because they are important scholarly tools that attempt to classify literature by discipline and fields of knowledge by compiling and categorizing individual publications of scholarly work into corpora. These two databases have important advantages over other compilations, including those for the social sciences [19]. We quickly realized that they would not allow us to complete the task, as their basic classification systems were inadequate.

The pursuit of knowledge in the academy is typically classified into three broad branches of knowledge: (biophysical/natural) science, social science, and arts and humanities. The metaphor of a branch presupposes separation—which precludes the possibility of overlapping domains, merging of streams of thought, and other metaphors applicable to transdisciplinarity—and thus has powerful implications for classification. These branches are reified as separate within the highest categories of many citation databases, including the very structure of WoS. By classifying individual publications as belonging to one branch of knowledge or another, entire transdisciplinary fields—like agroecology—are broken into two parts; if one works at the intersection of where these databases are divided, one can have a hard time finding and assessing all of the relevant literature. Understanding how a particular publication that falls only within the WoS science category engages with the social sciences is not possible with its current tools. Scopus has a more inclusive approach since it is a single corpus, yet filtering by the subject area "social science" leaves out a great deal of work published in biophysical and transdisciplinary journals that engage the social sciences.

Thus, transdisciplinary fields of knowledge, especially those that span the "branches" of biophysical and social sciences, present a classification problem for Western ontologies applied to academic research. Assessing how various fields interact—such as our goal of trying to determine the contributions of social sciences to the interdisciplinary field of agroecology—is a considerable challenge with current database classification systems and tools.

The transdisciplinary nature of agroecology meant we needed a new classification system to better discern the engagement of individual agroecology publications with the social sciences. This paper explains the method (Fig 1) we developed—the script-expert adaptive classification (SEAC) method—in depth (and differentiated into four phases) and compares its results with similar types of classification efforts by the two main multidisciplinary bibliographic databases, Scopus and WoS. We gathered a large corpus of peer-reviewed scientific literature that uses the term "agroecolog*" or "agro-ecolog*" in its title, abstract, and/or keywords. Our corpus, which we call the Agroecology Corpus 2019, contains 12,398 sources from peer-reviewed journal articles, books, and book chapters (S1 Dataset). We decided to develop an expert-informed semi-automated process of categorizing this corpus that was largely beyond direct human capacity. Utilizing a Python script we developed (**escholarship.org/uc/item/5hb78383**), we differentiated the literature into the three categories: not engaged with social science (NESS), engaged with social science (ESS), and uncertain engagement with social science (UESS). We then separated the Agroecology Corpus 2019 into three corpora for later analysis: agroecology engaged with social sciences (AESS, with 3,125 publications), agroecology not engaged with social sciences (ANESS, with 7,039 publications), and agroecology with uncertain engagement with social science (AUESS, with 2,234 publications).

Briefly, the process involved creating a list of 175 terms—such as "social," "feminis*" (to capture feminist and feminism, see Table 2 below), and "Polanyi"—that would help categorize the literature into different levels of engagement with the social sciences. We then created a multi-step filter to identify publications engaged with the social sciences. We iteratively tested it on a subsample of the literature that we had manually classified. Once we had a good working classification system for the manually-categorized subsample, we used the Python script to differentiate the larger corpus using the stepwise equation. The SEAC method allows us to categorize a large body of literature as engaged with the social sciences more accurately than by using common citation databases tools such as disciplinary and subject categories applied to journals and/or individual publications, and author institutional and departmental affiliation. Unlike previous tools, the SEAC method uses low computing power and is highly adaptable to different academic contexts. We are excited to share it since it could easily be applied further within agroecology, and adapted to other transdisciplinary fields and disciplines.

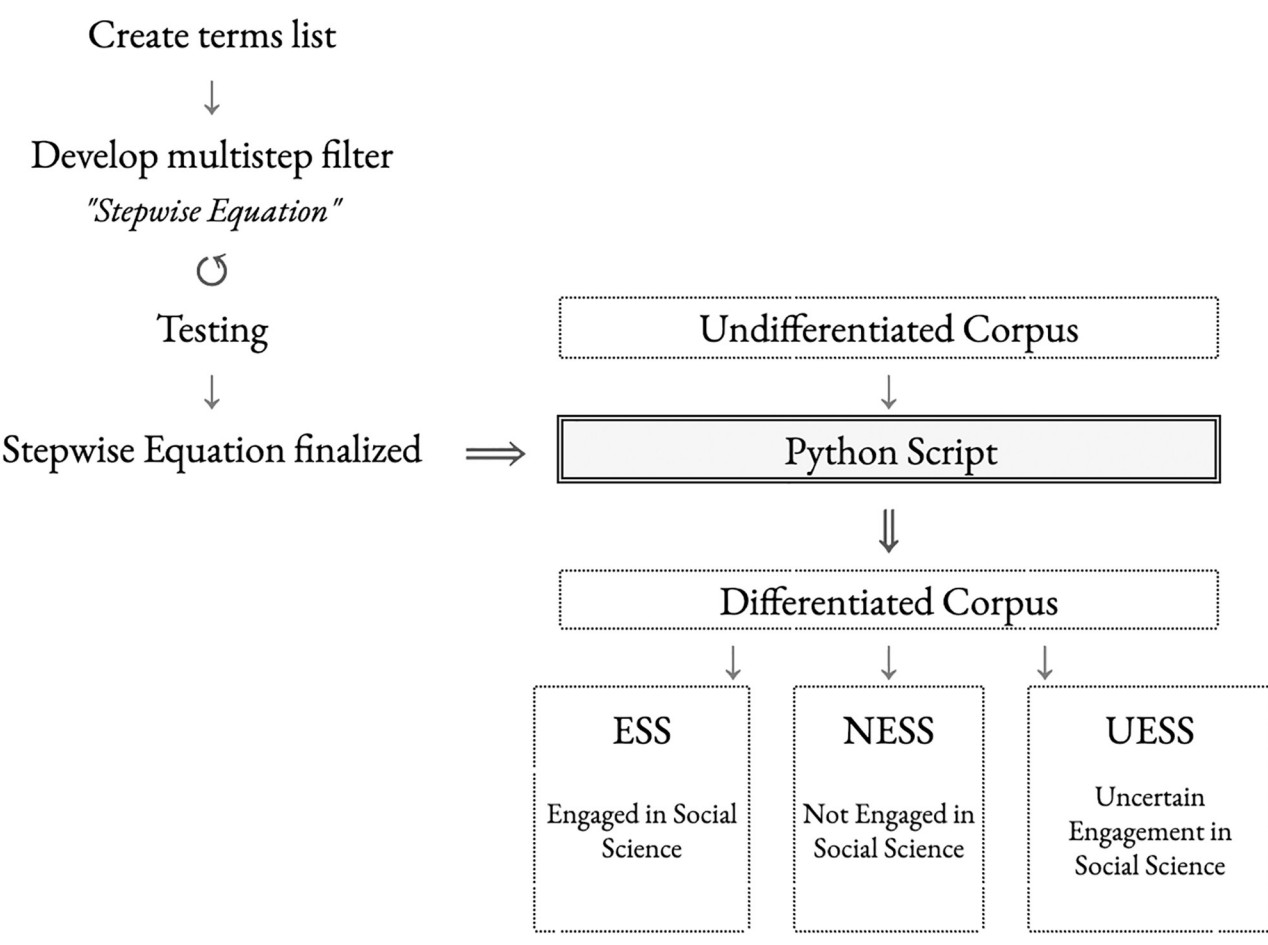

**Fig 1. An overview of the SEAC method.**

Next, we will examine relevant reviews of agroecology and the social sciences, and the bibliometric tools available for classifying transdisciplinary literature's engagement with the social sciences.

## Background

This section reviews two relevant areas of literature: 1) existing bibliometric tools that allow for identification of social science engagement within transdisciplinary literature, and 2) reviews of agroecology that examine its engagement with the social sciences.

**Existing tools for classifying social science engagement within in a transdisciplinary corpus.** The scale and scope of the literature in the environmental borderlands continues to expand. While publications exist showing how the social sciences can and do contribute to agricultural sustainability and agroecology [18] and environmental conservation efforts [9, 20], we were unable to find precise bibliometric tools to examine the engagement between the social sciences and biophysical sciences, including within transdisciplinary fields like agroecology.

WoS and Scopus have been critiqued for inadequately covering social science literature because of their tendency to misclassify it [21] and because of their strong emphasis on journal articles [19, 22]. Hicks [23] notes that one study found that biophysical scientists' publish 85%

of their output in journal and conference papers, while these venues account for 61% for social scientists' publications. Google Scholar has been presented as an alternative, yet lacks internal quality control and information on updates, and often presents incomplete publication information [24, 25].

Typically, researchers use the subject areas of Scopus to limit their findings to particular branches and/or fields of knowledge, e.g., [4]. A publication's subject area is assigned by the bibliographic database which indexes it, and is often determined by the subject area of the journal where it is published. The accuracy of such journal-level classification approaches has been an ongoing issue of concern and seriously called into question [26–29], especially for non-specialized, multidisciplinary journals which are growing in popularity. One recent study reports that journal-level classification has the potential to misclassify half of all papers [28] since articles were often not contributed from the journals' discipline. Meanwhile, the classification approaches of two of the most widely used databases, Scopus and WoS, both depend almost entirely on journal-level classification. This classification system has been scrutinized for being "too broad to adequately capture the more complex, fine-grained cognitive reality" [30]. Yet, these important multidisciplinary databases, and their questionable subject categories, are increasingly used in bibliometric studies such as global maps of science [31–33] and the accuracy of their classification system can have a substantial impact on the results.

It is no surprise that for transdisciplinary fields, some of the major categorization tools found within multidisciplinary bibliographic databases such as WoS and Scopus—e.g., journal subject categories, author institution, and department, and disciplinary or subject categories (such as the subject area of Scopus, e.g. "social science")—do not adequately identify texts as belonging to a transdisciplinary field, nor do they help identify the relative disciplinary contributions, let alone the appropriate branches of knowledge, within the source. When searching for publications that use the term "agoecolog*" or "agro-ecolog*" and restricting the search to the "social science" subject area (Scopus) or the "social science citation index" (WoS-SSCI), we find that at least 15% of the resulting papers are biophysical in orientation and do not engage with the social sciences (i.e., false positives). For example, the article titled "Agroecological management improves ecosystem services in almond orchards within one year" [34] is categorized as Social Science by Scopus *and* WoS, yet, as the title implies, has no engagement with any social science. Since we do not consider a 15% error rate for false positives to be adequate, an additional method is needed, as ready-made search engine tools are not yet up to the task.

The search for an adequate solution for classification of scientific publications has been intensifying as researchers explore ways to improve these classification systems. More granular classification approaches are continuously being proposed and evaluated. Machine learning has been the predominant approach to tackle this issue. With machine learning algorithms, computer assisted content analysis or automated text analysis can occur at very large scales. In the social sciences, developments include classifying articles into more fine-grained social science research disciplines and subdisciplines [30]; cluster analysis for subject category classification at higher resolutions to identify interrelationships based on citation networks [35]; replacing human annotation with machine learning to classify topics and subcategories, and to predict themes within social science documents [36]; text mining for qualitative text and content analysis [37]; and more generally classifying articles by clustering documents in common such as through references and textual data. Many advocate for article-based classifications that are better able to preserve the disciplinary differences that exist in individual articles, regardless of the journal they are published in [29, 30, 38]. Approaches that use primarily text classification (utilizing the title, abstract, full text, etc) are less common than those that use reference/publication data to identify publication relations. Yet, textual approaches can be more

sensitive and flexible [30] and thus are appropriate for transdisciplinary disciplines which are in constant flux, such as agroecology.

While these advances are promising, none have yet been developed into tools that could out-of-the-box allow us to examine the engagement between the social sciences and biophysical sciences within a transdisciplinary field like agroecology. Machine learning can require large computing power and advanced programming skills. Our goal was to create an easily reproducible method which requires low computing power and utilizes simple word processing tools and freely available software. We created a text classification approach (The SEAC method) that allowed us to adequately classify publications which are engaged and not engaged in social science within the field of agroecology, allowing us to conduct a much more accurate bibliometric analysis of the field.

**Reviews of agroecology's engagement with social science.** Previous bibliometric reviews of agroecology engaged with, or noted the existence of 141 sources in WoS and 208 in CAB in 2007 [17], 711 sources from Scopus complemented by university library portals and Google Scholar in 2008 [39], 574 European sources in Scopus in 2017 [40] and between 3,277 and 5,568 sources from WoS in 2020 [8]. Together these reflect the ever-increasing surge of literature using the term "agroecology" and its variations.

Two publications examine the history of the engagement of agroecology with the social sciences. Hecht [41] describes the history of agroecological thought beginning with colonization which led to the dislocation of local knowledge within agriculture. She identifies the views and fields that are central to agroecology including agricultural sciences, environmentalism, ecology, indigenous production systems, and development studies, and identifies the analytical tools that influenced the social dimensions of agroecology [41]. Sevilla Guzmán and Woodhouse [10] reveal the social foundations of agroecological thought. They articulate periods of social theory and discourse that gave way to agroecology (Marxism, modernist development, dependency and underdevelopment, peasant studies (1940–1990), post-development (1980 on), and environmental social theory (1980 on). Specifically, they identify the origins of agroecology in Marxist and libertarian social thought which supported social mobilization, reifying the indivisibility of science, movement, and practice.

Three reviews of agroecology that touch on engagement with social science utilize a bibliometric review approach to systematically analyze the field quantitatively and comprehensively [8, 40, 42]. While they did not conduct a close examination of the social science, Dalgaard et al. [42] searched for "agroecology" in a natural science database (ISIS-SCI), a social science database (SSCI), and an economics database (EconLit), and then looked at the overlap between those publications. While they did not specify the number of sources found, they stated that "66% of the references were only found in the natural science databases, with 13% only in a social science database, and 5% only in economic literature. No references were in the databases from all three fields of science" [42]. They did not investigate whether these results were a function of the databases, rather than an accurate reflection of the interdisciplinarity in the field. Gallardo-Lopez et al. [40] reviewed 429 agroecology papers based on research in Latin America and the Caribbean to identify the research methods, regions studied, and factors of analysis of agroecology (physical-biological, social, cultural, economic and political). They conclude that while agroecology in the region is rooted in social movements, conventional research methods (survey research, structured interviews, field studies and field experiments) still predominate scientific output and that research on the biophysical aspects of agroecology is the most common.

Most recently, Mason et al. [8] conducted a broad and systematic review of the field using bibliographic coupling networks to identify research communities and research fronts within agroecology. They then overlapped a portion of the cited references with the 250 WoS subject categories to identify the areas of knowledge from which agroecology draws, looking

specifically for integration with the social sciences. They identify multiple social science research fronts including agroecology education, peasant farming and industrialization, and food sovereignty and food systems. They conclude that agroecology is becoming "a very large, connected core" [8] which as a whole mainly draws from the natural sciences and much less, but increasingly so, from a wide range of social science, including development studies, environmental studies, anthropology, geography, sociology, and economics.

While Dalgaard et al. [42] and in particular Mason et al. [8] are excellent contributions to our understanding of the panorama of agroecological scholarship, they assume that scholarly databases adequately categorize journal articles and publications by branch, field, and discipline, even when they appear in multi-, inter-, and transdisciplinary journals. These journals can be assigned unambiguous categories, yet they contain large numbers of publications which are heterogeneous in their disciplinary engagements and thus difficult to accurately bound. In our experience looking at the issue in depth, bibliographic databases do not very accurately identify the composite fields and disciplines of transdisciplinary scholarship.

The rest of the paper is organized as follows: We next describe the SEAC method in great detail to demonstrate it as a replicable method. Then, we analyze the efficacy of the SEAC method compared to commonly available tools. Finally, we then conclude the paper by reflecting on the methods' strengths and weaknesses.

## Methods

The development of the SEAC method involved four phases (Fig 2): 1) gathering literature, 2) identifying social science terms that could be used to identify agroecology engaged with social

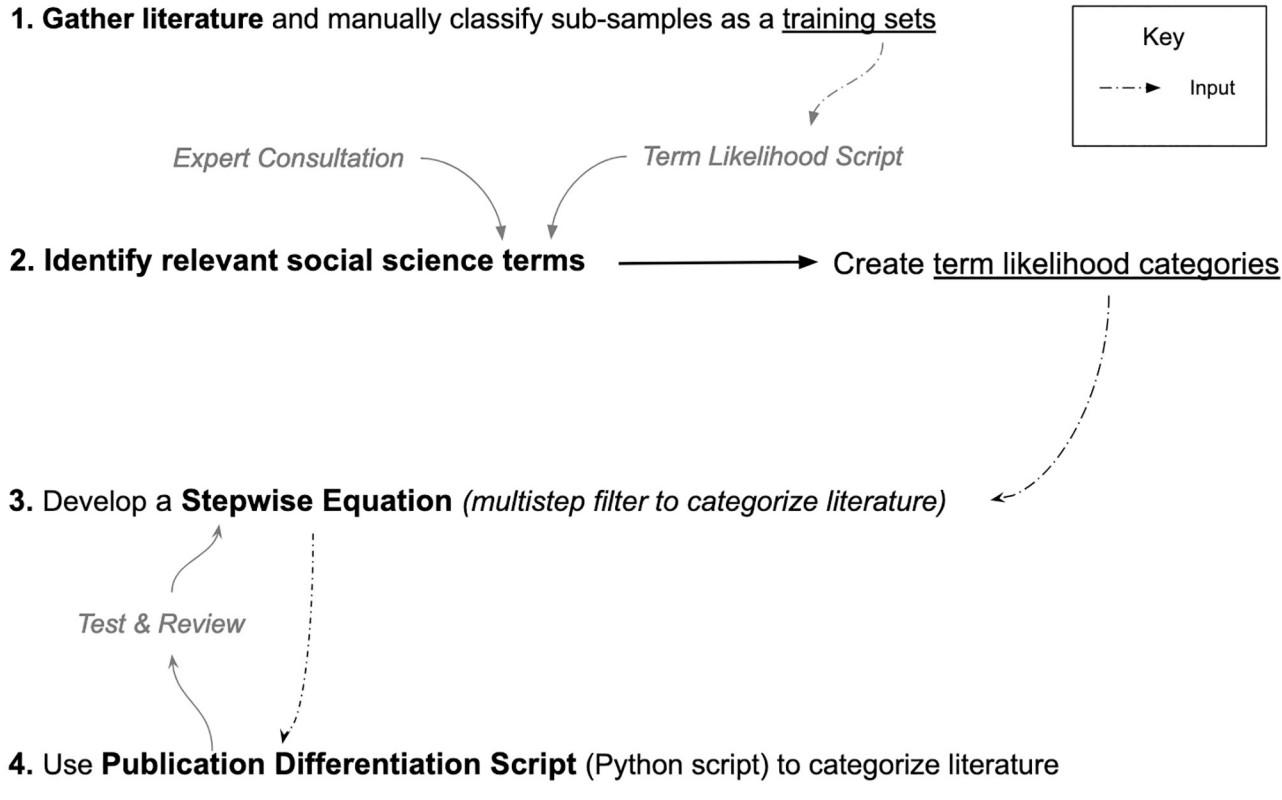

**Fig 2. Developing the SEAC method.**

**Table 1. An example of a publication differentiation script result.**

| Title Abstract | Seeding agroecology through new farmer training in Canada: knowledge, practice, and relational identities | Term Likelihood | Variety | Frequency | Terms |
|---|---|---|---|---|---|
| As a concept, agroecology emphasises the interweaving of scientific and traditional ecological knowledge and is evolving in conjunction with farmer-led social movements from around the world addressing the health, equity and ecological sustainability of food systems. in canada, many new agroecological farmers come from non-farming backgrounds and are finding limited training opportunities and support structures. while there is a growing literature on the evolution of agroecology, there is limited research on the existence and impact of training programmes on the subject-formation of new farmers. in this paper, we consider the subject-formation of new agroecological farmers through a case study of the everdale community learning centre, one of canada's only agroecological farm schools. in particular, we explore how the knowledge, practice, and relational identities of participating graduates are informed by and build on the science, practice, and movement of agroecology. drawing on a survey and interviews with past participants, we found that everdale's education programme contributes to an agroecological subject-formation by promoting the co-creation of place-based agricultural knowledge// teaching the complexities of agroecology practice through both experiential and theoretical training// and, building a supportive community of peers. we conclude with reflections on ways to encourage a greater diversity of new farmer entrants and opportunities to support training programme graduates in establishing successful farms. these findings provide insight into developing new agroecological farmers and supporting the growing agroecological movement in canada. | | **DSS** *Definitely Social Science* | **1** | **2** | social movement |
| | | **LSS** *Likely Social Science* | **8** | **8** | education, experiential, interview, learn, participa, school, social, theoretic |
| | | **PSS** *Possibly Social Science* | **4** | **10** | training, case stud, food system, movement |
| | | **MSS** *Maybe Social Science* | **3** | **18** | knowledge, farmer, sustainab |
| **Keywords** | agroecology;canada;farmer knowledge;farmer training;food movement;new farmers;farmers knowledge;seeding;smallholder; teaching;training | Total Score | **89** | | |
| | | **Result:** | Engaged in Social Science | | |

science, 3) developing a stepwise equation for categorizing publications, and 4) categorizing the literature into the three discrete categories: a) *not engaged with social science* (NESS), b) *engaged with social science* (ESS), c) *uncertain engagement with social science* (UESS). We describe each of the phases in detail below to allow others to apply the method or a modified version of it.

We start by demonstrating the results of the SEAC method on a single publication (Table 1). The method uses 175 social-science related search terms grouped into four term likelihood categories—definitely social science (DSS, 100% likelihood), likely social science (LSS, 99% to 85% likelihood), possibly social science (PSS, 84% to 75% likelihood), and maybe social science (MSS, 74% to 65% likelihood)—and then finds instances of each of these terms in each publication's title, abstract, and keywords. The script then counts the variety and frequency of each term within the four term likelihood categories. For example, the script found one DSS term—social movement—and it appeared twice (top row of Table 1). For the MSS terms (penultimate row of Table 1), it found three—knowledge, farmer, and sustainable—and these appeared 18 times. In addition to the variety and frequency counts for the four term likelihood categories, we also created a total score category that weighed each of the term likelihood categories differently (heavy weights for DSS, progressively dropping to light weights for MSS, with details in Phase 3 below). We then developed a stepwise equation to find thresholds that would allow us to determine whether a publication was engaged with social science (ESS) or not (NESS). In the case of the publication in Table 1, there are a considerable number of social science terms found, which means the script classifies it as ESS. We now turn to explaining each phase of the method in detail.

## Phase one: Gathering literature

In early 2020, we gathered literature on agroecology from different citation databases using standardized search criteria. We gathered all peer-reviewed journal articles, book chapters, and books published from any year until and including 2019 that contained the term "agro-ecology" within the keywords, title, and abstract. This was operationalized as the specific search phrase "agroecolog** OR agro-ecolog*" with the asterisks as wildcards that could catch either "agroecology" or "agroecological" and the "OR" term allowing us to find the spellings with and without a dash (both remain common). We performed the search in 1) the two most important multidisciplinary bibliographic databases, Scopus and WoS (the WoS Core Collection and the Social Citation Index (SSCI)); 2) two specific citation databases focused on agriculture, Agricola and CAB Direct; and 3) one specific database on economics, EconLit. We excluded Google Scholar from our citation database selection because of its lack of internal quality control [25] and its lack of information on how it is updated [24]. All publications from each search were downloaded, compiled, and saved in one library in the citation management software *Endnote*. The outcome was what we refer to as the *Draft Agroecology Corpus*.

We then cleaned the Draft Agroecology Corpus. This involved four processes, each described below: we removed publications without abstracts, adjusted metadata as appropriate, removed duplicates, and removed false positives (publications which a database had labeled with agroecology, but which did not originally refer to agroecology, largely because of database-created addendums to abstracts).

We decided to remove publications without abstracts because abstracts provide valuable information for accurate categorization. While our first attempts at categorization included these publications, we found that without abstracts, there was not enough information to accurately determine the level of engagement with social science. Removing publications without abstracts affected 592 publications; however, important books were retained, as a book summary appears in the abstract field of some books.

In handling metadata, we were careful that metadata was imported in the same fields for each citation database. For example, since the treatment of author keywords varies greatly by citation database, we made sure that, when available, author keywords were imported into the same field ID for each citation database used. We also made sure book chapters were imported as a reference type "book chapter" rather than a "journal article" reference type. We also noticed that journal titles are sometimes written differently depending on the database, so we adjusted journal titles so that each journal had a consistently spelled title. This also helped better identify duplicates and to later analyze the results.

To remove duplicates, we first used *EndNote*'s duplicate identifier tool, followed by a similar tool in *Google Sheets*. We also manually found duplicates by sorting the database on different fields (especially article/book/chapter title) and looking for two or more with the same data. When merging duplicates, we left the publication from the citation database that was the more preferred database based on completeness and accuracy of publication record. We decided on a hierarchy of more preferred databases (Scopus > WoS > EconLit > Agricola > CAB) by determining which had more consistently completed fields and greater accuracy with the way metadata was pulled into *EndNote*. For example, with duplicates from Scopus and WoS, we always kept the Scopus version of the publication.

Although we searched only for "agroecology" (and its three other iterations) in titles, abstracts, and keywords of peer-reviewed literature (journal articles, books, and book chapters), upon examining the resulting database we found that some publications did not meet these criteria. We first found these false positives manually, then determined how to find them systematically. First, some publications were not of journal articles, books, or book chapters.

Thus, we removed magazine articles and other non-peer-reviewed literature. Second, some false positives were publications that did not contain the words "agroecology," "agroecological," "agro-ecology," or "agro-ecological" in the original abstract, title, or keywords. These false positives occurred because some databases amend these fields with their own metadata. We found this through preliminary analyses, seeing that some journals and institutions had an outsized number of publications. For example, the journal *Chinese Geographical Science* yielded many false positives because some databases appended abstracts with the affiliation of the authors at the Northeast Institute of Geography and Agroecology. When these publications did not have any other mention of agroecology, they were deemed false positives and removed.

After the cleaning processes noted above, we had 12,398 peer-reviewed publications with the terms "agroecology," "agroecological," "agro-ecology," or "agro-ecological" in the original title, original abstract, and/or author keywords. These comprise what we call the *Agroecology Corpus 2019* since it is up to date as of the end of 2019 (S1 Dataset).

Agroecology Corpus 2019 contains a number of publications that, although they use the term "agroecology" and/or its iterations, may not be engaging with the field of agroecology. Some of these publications can be identified by selecting those that *only* use "agroecological" or "agro-ecological" followed by a geographical term (especially "zones") or physical indicator as was done by Mason et al. [8]. Rather than remove all of these publications as Mason et al. [8] did for their analysis, we decided to keep them for two reasons. First, we decided to defer the refinement of the *type of engagement with agroecology* to another round of analysis, as it would exceed the word limits of a traditional article. Second, since there is considerable contestation of what agroecology is as a field [11, 14], we left them in for those seeking to understand potentially all aspects of the discipline, including if future analysts decide that papers discussing "agroecological zones" are indeed part of agroecology. These are potentially intriguing to consider; an example is the earliest ESS publication in our database, titled "India's new agricultural development: the case of a conflict between agricultural growth and social equity" [43]. This article only uses the term "agro-ecological conditions" in the abstract, yet it examines income disparities, conflicting goals, and agrarian unrest as a result of agricultural development. We wanted to retain such historical examples in our analysis.

## Phase two: Identifying social science terms

We next consulted experts in the social science disciplines at the interface with agroecology to gather a preliminary list of terms that are most likely to categorize sources vis-à-vis their engagement with the social sciences. We consulted with fellow political ecologists of food and agriculture at UC Davis to help us come up with an initial set of 97 terms that could be used to identify whether a publication is engaged with the social sciences. We then organized the terms based on our best estimate of the probability that the term would accurately categorize the publication as ESS or NESS. To do this we placed the social science terms into four *term likelihood categories*, initially based on our prediction about how likely the words would be to identify ESS publications: *definitely social science* (DSS) for terms used only in social science, such as "Gramsci" and "actor network theory;" *likely social science* (LSS) for terms very likely to be used only in social science, such as "ethnography;" *possibly social science* (PSS) for terms very commonly used in social science but also used sometimes in the biophysical sciences, such as "food security;" and *maybe social science* (MSS) for terms slightly more commonly used in the social sciences than the biophysical sciences, such as "sustainable." We iteratively tested these terms' capabilities of identifying ESS publications, thereby adapting the term likelihood categories to the results, as described below. Through this process we also identified

**Table 2. 175 social science terms within the four term likelihood categories.**

| | Definitely Social Science (DSS) | Likely Social Science (LSS) | Possibly Social Science (PSS) | Maybe Social Science (MSS) |
|---|---|---|---|---|
| *Likelihood term will correctly identify an ESS publication* | 100% | 99% to 85% | 84% to 75% | 74% to 65% |
| | racis✗ -in. (racism, racist) -ex. (anthracis); sociolog° -in. (sociology, sociological) -ex. (phytosociology, plant sociology); actor network theory✗; agrarian studies✗; decoloniality✗; decolonization✗; discourse analysis ✗; discourse✗; embeddedness✗; enthnograph✗ -in. (ethnography, ethnographic); epistemolog✗ -in. (epistimology, epistemological); ethnic studies✗; feminis✗; foucaul✗; gramsci✗; grounded theory✗; hegemon✗; human rights✗; imperialism✗; indigeneity✗; international studies✗; intersectional✗ -in. (intersectionality); justice✗; latour✗; marx✗; oppression✗; peasant studies ✗; polanyi✗; political ecolog✗; political econom✗; political science ✗; political-ecolog✗; political-econom✗; positionality✗; queer✗; science and technology studies✗; science studies✗; social movement✗; social science✗; sovereign° | -econo✗; agraria -in. (agrarian); alternative food network✗; anthropolog✗; argue° -in. (argued, argues); attitude; autonomy✗; capitalis -in. (capitalism, capitalist); collaborat ° -in. (collaborate, collaboration, collaborative); commodity chain✗; commodity-chain✗; cooperat° -in. (cooperative, cooperation); cost-benefit✗; critical consciousness✗; design process✗; direct market✗; disciplinary -in. (multidisciplinary, interdisciplinary); document analysis ✗; education -in. (educational); empower° -in. (empowered, empowerment); engage° -in. (engagement) -ex. (engaged); engaged°; ethno✗ -in. (ethnobotany, ethnography, ethnographic); experiential°; focus group✗; gender°; illustrate -in. (illustrates, illustrated); implement -ex. (implements, implemented and implementation); institution -in. (institutional, institutionalization); interview -in. (interviewed); labour°; learn° -in. (learned, learning); marginalization✗; marketing°; ontolog✗; participa -in. (participate, participation, participatory); peasant°; pedagog✗ -in. (pedagogy, pedagogical); polic -in. (policies, policy); politic -in. (political, politics); reform° -in. (reformation); ricardian✗; ricardo✗; school°; settlement°; social -in. (socially); socio -in. (socioeconomic, sociopolitical, etc) -ex. (phytosociology); socioeconomic°; solidarity✗; supply chain✗; supply-chain✗; theoretic°; tradition -ex. (traditional); value chain✗; value-chain✗ | afn ✗; corporat✗ -in. (corporate) -ex. (incorporate); power -in. (powerful) -ex. (empower, empowered, empowerment); race ✗ -ex. (landrace); sts ✗; case stud✗; collective° -in. (collectively); commod✗ -in. (commodity, commodities); commons✗; communicat° -in. (communicating, communication); econo -in. (econometric, economic, economical, economically, economics, economist, economy); extension -ex. (extensions); financ ° -in. (financial); food security✗; food system✗; govern -ex. (governing, governance, government, governmental); governance°; histor; homegarden✗; household; income; innovation°; intervention; livelihood✗; market -ex. (marketing); movement°; narrative✗; organization -in. (organizational); organizational°; paradigm°; perspective; poverty°; price; question -in. (questionnaire); rural✗; systemic✗; technol technology, technological -in. (technology, technological); territorial°; training°; transform -in. (transformation, transformative, transforming, transformed); urban -ex. (urbanization, urbanizing); vulnerab° -in. (vulnerable, vulnerability) | knowledge -ex. (acknowledge); labor -ex. (collaborate); adopting °; adoption°; communal✗; decision; dialogue✗; discussion°; embedded ✗; family; farmer; garden° -in. (gardening); governmental°; implemented✗; implementing°; initiative°; language✗; law; legal°; modern -ex. (modernization, modernizing); moderniz° -in. (modernization, modernizing); people°; perceive✗ -in. (perceived); perception✗; practitioner°; researcher°; security -in. (insecurity); societ° -in. (society, societies, societal); stakeholder°; strateg° -in. (strategy, strategies); sufficiency✗; sustainab -in. (sustainable, sustainability); systema° -in. (systematize, systematic, systematically); think° -in. (rethink, rethinking, thinking); tourism°; tradeoff✗; urbaniz° -in. (urbanizatio, urbanizing); village° |

✗ placement based on limited data ° placement based on no data -in. = "includes" -ex. = "excludes"

**Terms eventually excluded** due to poor correlation or too many false positives identified: *agency; agri-food; agriculturally; agrifood; agro-food; agrofood; colonization; community; cooperatives; critical; cultur; cultural; cultural ecology; democra; development; farming; food; health; human; indigenous; network; nutrition; profit; public; qualitative; rights; smallholder; trade*

additional terms that appeared more frequently in ESS publications and much less frequently (or not at all) in NESS publications.

Table 2 shows the final version of all 175 words in the four categories, as well as a footnote with those terms initially tried but eventually excluded. We used specific cutoffs for each of the term likelihood categories: DSS (100%), LSS (99% to 85%), PSS (84% to 75%), and MSS (74%

to 65%), with the percentages representing the likelihood that publications using the term were manually classified as ESS versus NESS.

After we created the first version of our term likelihood categories, we created four training sets (subsets of the Agroecology Corpus 2019) for working on the script to see how accurate it was. The first three sets were based on specific journals that had the highest number of publications within the journal categories that we created (social science journals, biophysical science journals, and mixed social science-biophysical science journals); categorization was based on reading journal mission statements. One training set was all papers from *Agriculture and Human Values* in the Agroecology Corpus 2019, a multidisciplinary social science journal; we used this as a training set because we could safely assume that the vast majority of the papers in the journal engaged with the social sciences, though we manually classified the publications to confirm. When we ran the script, looked at the results, and iteratively improved it, we saw that the script was usefully differentiating papers within journals. For example, in *Agriculture and Human Values*, we saw that one paper had a very low score, suggesting it was not social science. Upon further examination, this paper was from Guthman (1998), with which we were familiar, and which we realized was an agroecological assessment of organic growers' practices in California, with an almost entirely biophysical science focus. Thus, we saw early on that the script was allowing for fine-grained differentiation based on engagement with social science.

Another training set was all papers from *Field Crops Research* in the Agroecology Corpus 2019, a multidisciplinary biophysical science journal; we used this as a training set because we could safely assume that few of the papers included in the journal engaged with the social sciences. The other journal-based set was all papers in the Agroecology Database from *Agricultural Systems*, a multidisciplinary journal that commonly has mixed biophysical and social science analyses, which would be helpful to see how the script handled a more mixed journal. The fourth set, which we call the *random training set*, was a randomly selected subset of 368 publications from the Agroecology Corpus 2019. Together we call these four training sets the *full training set* (Table 3).

By reading the titles and abstracts of each publication in the four training sets, Pinzón and Galt manually classified each into the three publication categories that we had hoped to categorize all publications into: "engaged with social science" (ESS), "partially engaged with social science" (PESS), and "not engaged with social science" (NESS). We ultimately were not able to adequately identify partial engagement (PESS) through the script, so this category was not used in the final classification results. Instead, we decided a category of uncertain engagement with social science (UESS) was a more accurate way forward, as described below. To make our manual determination, if reading the title and abstract for a publication was insufficient for determination, we skimmed the full text with particular attention to the methods section. For publications we were uncertain about, we discussed in detail until we reached a consensus.

**Table 3. The training set.**

| | Total Publications | ——ESS—— | | ——NESS—— | | ——PESS—— | |
|---|---|---|---|---|---|---|---|
| | | n | % | n | % | n | % |
| **Journal-based training sets**: | | | | | | | |
| *Agriculture and Human Values, Field Crops Research, Agricultural Systems* | | | | | | | |
| | 262 | 145 | 55% | 90 | 34% | 27 | 10% |
| **Random Training Set:** | | | | | | | |
| a random subset of the larger corpus | | | | | | | |
| | 368 | 121 | 33% | 200 | 54% | 47 | 13% |
| **Full Training Set** | **630** | 266 | 42% | 290 | 46% | 74 | 12% |

The full training set (the four training datasets together, Table 3) had 630 publications, with 266 ESS publications, 74 PESS publications, and 290 NESS publications.

We then developed and ran a *term frequency script* on the 630 publications within the full training set to determine in which of the four term likelihood categories (DSS, LSS, PSS, and MSS) each of our search terms should fall. The script searches for all of our previously-identified social science terms (Table 2) in the abstract, title, and keywords of each publication, and then identifies how many publications contain the term, thereby allowing us to know the total number of ESS and NESS publications in the full training set for each term. We used these publication counts and the probability that a term would be found in ESS publications compared to NESS publications to examine the differences in the occurrences of each term between the manually-classified ESS and NESS publications. The initial probability ($p_i$) calculation was:

$$p_i = ESS/(ESS + NESS)$$

Placing a term within a specific term likelihood category (DSS, LSS, PSS, and MSS, each with their probability range) was a highly iterative and adaptive process. The random training set is the most representative of the full dataset; thus, when sufficient information was present (ESS+NESS for a term $\geq 20$), we decided to use the probability calculated with this dataset (with an error term, described below). However, if that dataset did not have enough information, we used additional information from the full training set. If ESS+NESS for a term in the 368-publication random training set was between 10 and 20, and the ESS+NESS for the same term in the full training set was $\geq 10$, we calculated the average of the two percentages and used that for the classification. If ESS+NESS for a term in the 368-publication random training set was $\leq 10$, we determined there was insufficient information to use the probability calculated (even with the error term). In these cases, we either 1) used our best estimate (e.g., "sociolog" as DSS, found in five ESS publications and zero NESS publications in the full training set), or 2) used the probability calculated from the full training set (also calculated with an error term, described below).

We assumed an error term for the probability calculation that was inversely proportional to the combined sum of 1) the total of the ESS publications in which the term was found and 2) the total of the NESS publications in which the term was found. We used the following equation to determine the error-adjusted probability ($p_e$) for each term:

$$p_e = (ESS - 1)/(ESS + NESS)$$

Adjusting for error was primarily to be conservative with the placement of terms generally, and more conservative with terms that had a high ratio of ESS to NESS but were only found in a small number of publications. Terms that both had a high ratio of ESS to NESS and were found in a large number of publications were thus much less affected. We applied this error calculation to the probability calculations for the full training dataset and random training dataset. This process can be illustrated with the example term "farmer." In the case of "farmer," 50 ESS publications and 16 NESS publications in the random training set had the term. Since ESS+NESS > 20, we went with the data from the random training set. Thus, we calculated that there was a 76% initial probability—calculated as $p_i$ = 50 / (50 + 16)—that the term would correctly identify an ESS publication (top section of Table 4). To add the error term, we calculated $p_e$ as follows:

$$p_e = (50 - 1)/66$$
$$p_e = 74.2\%$$

Table 4. Term frequency script results for the term "farmer".

| Term | ESS Publications | NESS Publications | TOTAL Publications | Probability Document is ESS (Pi) | Probability with error calculation (Pe) | Term Likelihood Category |
|------|------|------|------|------|------|------|
| [Random Training Set] N = 368 publications (121 ESS; 200 NESS) | | | | | | |
| farmer | 50 | 16 | 66 | 76% | 74% | MSS |
| [Full Training Set] N = 630 publications (266 ESS; 290 NESS) | | | | | | |
| farmer | 144 | 40 | 184 | 78% | 78% | PSS |

Thus, while the initial probability calculation would have placed it in PSS (since it was above 75%), adding the error term resulted in a more conservative placement of "farmer" into MSS (below 75%).

By applying the script and sifting through the results, we also tested the specific terms in each of the term likelihood categories (Table 2), adding and dropping terms as we proceeded. Applying the script to the full training set allowed us to discard terms that were suggested during expert consultation but that did not clearly distinguish ESS and NESS publications (see Table 2's footnote about terms eventually excluded).

Applying the script iteratively also allowed us to find new terms not considered during expert consultation in two ways. First, when publications manually classified as ESS publications did not have any identified terms associated with them, we read their titles and abstracts to see what other terms might have been useful in classifying the publication, then added these terms to the script to see if they were useful for other publications. We kept them if they had probabilities in the MSS category or above ($\geq$65%). Second, we looked for terms that appeared frequently in ESS publications and far less frequently in NESS publications to identify unexpected terms not identified by expert consultation. Examples of these added terms include "adoption," "farmer," "perception," "tourism," "village," "discussion," and "think" (all MSS terms).

Using the results from the process described above, and later verified in Phase 3 (below), we organized all the terms into their final locations in the four term likelihood categories (Table 2).

## Phase three: Developing a stepwise equation for categorizing publications

Once the term likelihood categories were finalized, we developed it into a *publication differentiation script* and ran the script on the random training set. The publication differentiation script is the central script used in the SEAC method. It is a script written in Python programming language that counts the occurrence of each term from the term likelihood categories (see Table 2) in the title, abstract, and keywords of each publication. Terms appearing in each of the four term likelihood categories (DSS, LSS, PSS, and MSS) are identified within a publication's title, abstract, and keywords, and each category (DSS, LSS, PSS, and MSS) is given a value for 1) the sum of the *frequencies* of all terms (how many times all the terms in each category appear) and 2) the sum of the *variety* of all terms (how many different terms in the category appear). In this way, each reference received two scores (frequency and variety) for each of the term likelihood categories and then a total score, for a total of nine scores (Table 1). The total score was calculated as follows, with different weightings for each of the term likelihood categories:

$$Total\ score = (DSS\ variety \times 10) + (LSS\ variety \times 8) + (PSS\ variety \times 3) + (MSS\ variety \times 1)$$

**Table 5. Stepwise equation.**

| | A publication is not engaged in social science (NESS) if ANY of the following are true: | | |
|---|---|---|---|
| | **Variety** | **Frequency** | |
| | **Number of unique terms** | **Frequency of appears of those terms** | |
| DSS | 0 | 0 | *or* |
| LSS | < 3 | < 4 | *or* |
| PSS | < 4 | < 8 | *or* |
| MSS | < 5 | < 8 | *or* |
| **Total Score** < **11** | | | |

DSS and LSS were weighted very heavily since terms in both categories are very likely to identify a publication engaged with social science without a high risk of false ESS positives. PSS and MSS are weighted considerably lower since they were much more likely to generate false ESS positives. These weights were developed upon systematically reviewing the values that the manually-categorized ESS, PESS, and NESS publications received across each of the nine scores. We only used variety in the total score, rather than variety and frequency, because we found that the occurrence of a term (variety) was a stronger signifier of ESS than the number of times the term was used. Frequency was however incorporated in the *stepwise equation* as seen below.

We developed a *stepwise equation* (Table 5) that took into account the different scores, and the thresholds within each that indicated whether a publication was ESS or NESS. We sought to determine the value at which each of the nine scores reliably identified publications as NESS or ESS. To do this, we reviewed the nine scores received by each of the 630 publications within our full training set. We looked for patterns that would allow us to clearly distinguish ESS from NESS, with special focus on cutoffs at which there were no more NESS, and the grey area of score thresholds where NESS, PESS, and ESS publication all appeared. While a formal fitting procedure could be used here, we opted for a less mathematically intensive approach. For example, we chose to designate publications with a cutoff of a total score less than 11 as NESS. We arrived at this number by finding where we could strike a balance between losing ESS from the ESS category (as false negatives) and incorrectly keeping NESS (as false positives) in the ESS category. In our testing data, only 28 NESS publications received a total score of 11 or greater and only 24 ESS publications had a score below 10; therefore we chose a score of 11 as a middle ground where that balance was struck. Through repeated selection and review, we sought a score threshold for each of the nine scores that resulted in identifying the least number of NESS possible and the largest number of ESS possible. We also tested and adjusted the Total Score formula in this phase, trying four different equations (largely with different weights) before settling on our final one described in Table 5 below.

To validate our stepwise equation we selected a *random validation set* of 100 publications (different from the random training set of 368). We manually classified it in the same manner described above and then subjected the validation set to the publication differentiation script. We then compared the results of the manual versus automated classification for the random validation set and used these results to adjust our stepwise equation and term likelihood categories as needed. For the latter, we continued to improve the location of our terms within the four categories by identifying terms that commonly led to false positives and/or false negatives. For example, we made certain that DSS (100% probability of correctly identifying an ESS publication) variety and frequency scores were zero for publications that were manually classified

as NESS; when a DSS term appeared for a NESS publication, the term was dropped down to the LSS list. Terms generating false positives were dropped down in the term likelihood categories; e.g., the term "govern," which appeared 10 times in false positives, was moved from LSS to PSS; similarly, the term "adoption," was dropped from LSS to MSS. Using the methods above we iteratively adjusted the *term likelihood categories* five times, going back through all of Phase 2 and Phase 3 to obtain the lowest number of false positives and false negatives possible for each of the terms. We sought to reduce the error by adjusting the placement of the terms within the terms likelihood categories in the publication differentiation script to create results more similar to the manual classification.

For those wishing to replicate the SEAC method, we would suggest simply using the stepwise equation directly as it appears here and would not recommend developing a new equation from scratch unless it was necessary or desired to improve performance and accuracy. However, future iterations of this process could likely be systematized with machine learning, optimization, or other data science methods.

## Phase four: Categorizing the literature

Adjusting the terms and the stepwise equation thresholds iteratively as described above produced a final term likelihood categories list (Table 2) and a final stepwise equation (Table 5) that we operationalized through the final publication differentiation script to determine whether a publication was potentially ESS or NESS. The first step of the final differentiation process was that any publication that met the criteria of the stepwise equation was labeled as NESS and placed into what we call the *Agroecology Not Engaged with Social Science Corpus* (ANESS Corpus), which contains 7,039 publications.

Next, we wanted to differentiate the remaining publications (i.e., all publications that were potentially ESS) and separate those that we were highly confident were ESS from those that we were not highly confident were ESS. When looking at our test results of false ESS positives and false ESS negatives, we found that all incorrectly classified publications scored at or below 24 in the Total Score. Since we wanted a group of ESS to be correctly categorized with high confidence, we decided to use a Total Score of greater than 24 to differentiate the *Agroecology Engaged with Social Science Corpus* (AESS Corpus), which contains 3,125 publications. This left the publications with a Total Score range of 11 to 24 in our *Agroecology with Uncertain Engagement with Social Science Corpus* (AUESS Corpus), which we decided would be our group of publications with uncertain engagement with social science (UESS). This "gray zone" ended up with 2,234 publications.

To test the SEAC method, we selected and classified a *random **testing** set* of 100 publications, independent of the sets used to train and validate our method. There is no consensus on the minimum number of publications needed to test the performance of classification filters [44]. We measured the performance of the method in terms of sensitivity and precision for both the AESS and ANESS corpora and the overall accuracy. Within the field of bibliometric classification, precision is the proportion of retrieved documents being relevant and sensitivity is the proportion of relevant documents being retrieved [44], and accuracy is the overall success of the classification method [45]. We measured the precision, sensitivity, and accuracy of the AESS and ANESS.

- Precision AESS = (True ESS Retrieved) /(All ESS Retrieved)

- Sensitivity AESS = (True ESS Retrieved)/(All ESS)

- Accuracy = (All True ESS+NESS Retrieved)/(All ESS+NESS Retrieved)

Given the difficulty in humans *and* machines classifying the PESS (see Table 6, g), we did not measure the performance of UESS as there were a number of publications which are either not easily classifiable without full texts, or are only marginally engaged in social science.

Overall, the SEAC method performs very well (Table 7) with an average of 90% precision, 87% sensitivity and 90% accuracy across two categories of AESS and NESS (Table 6). In a separate study, a comparable automated textual classification method produced 89.6% precision, 89.4% sensitivity and 90.7% accuracy when using a boolean mixed filter [45]. In yet another study, a comparable machine learning method used to classify sociology publications achieved 81% accuracy [46]. The sensitivity of the AESS corpus is relatively low because we opted for a higher precision with as few false positives as possible even if it meant losing potential ESS publications.

The final three corpora produced by the SEAC method included AESS, with 3,125 publications; ANESS, with 7,039 publications; and AUESS, with 2,234 publications; 12,398 publications total. Fig 3 shows the probable composition of the three final corpora. We used the results of the validation and testing data to calculate the composition of the corpora by ESS, PESS, and NESS publications. We estimate that the AESS Corpus is 92% ESS publications, 9% PESS publications, and 0% NESS publications, making it entirely ESS and PESS. The ANESS Corpus is estimated to be 92% NESS publications, 7% PESS publications, and 1% ESS publications. Thus, we use AESS Corpus as the group of ESS publications for the analysis below, and compare it to ANESS Corpus as the group of NESS publications. For most of our analysis, we leave out UESS Corpus since it is a fairly even mix of the three (50% ESS, 29% PESS, and 21% NESS).

## Findings: Comparing databases and methods

We now compare the results of the SEAC method to those of the two main composite citation databases, Scopus and WoS, in their ability to identify agroecology engaged with social science. While we made direct comparisons, we should note that the SEAC method did not force a binary decision, unlike Scopus and WoS. In other words, the SEAC method contains the UESS (uncertain engagement with social science) category for unknowns, while Scopus and WoS both require a specific publication to be labeled as social science or not social science. Nevertheless, we can directly compare the results of their identification of social science publications.

Table 6 above shows that the SEAC method produced a higher overall accuracy (90%) than Scopus and WoS (66% and 67%, respectively). This can be mitigated by gathering the ESS from *both* Scopus and WoS and combining the results, which produces an accuracy of 70%. When combined, Scopus and WoS have a significantly higher ESS sensitivity (89.6%) than the SEAC method (77%). However, within those "ESS publications", 15% are NESS and 12% PESS (or 27% false ESS positives). The SEAC method, though less sensitive as we have calibrated it, is much more precise. Using a non-binary approach allows the SEAC method to produce a AESS corpus which has nearly 0% NESS and less than 10% PESS (Table 6, Fig 3). An additional strength is that, by utilizing multiple databases beyond Scopus and WoS, an additional 3,622 publications which were not indexed in Scopus or WoS were found by the SEAC method. Its accuracy and precision alone makes the SEAC method a powerful social science classification tool when compared to currently available multidisciplinary bibliographic databases.

To examine the differences between the classification systems more deeply, we compared five key journals for agroecology publications. The top rows of Table 8 show the total number of agroecology publications found in each database, and how those databases classify the publications according to whether they are social science (SS) (for the SEAC method, we only use

**Table 6. Performance of the SEAC method.**

| | Total in AESS bin | | | Total in ANESS bin | | | Number of unclassifiable publications | AESS Performance | | ANESS Performance | | |
| --- | --- | --- | --- | --- | --- | --- | --- | --- | --- | --- | --- | --- |
| | Number of ESS retrieved | Number of erroneous classifications | Number of ESS not retrieved in the ESS bin | Number of NESS retrieved | Number of erroneous classifications | Number of NESS not retrieved in the NESS bin | | *how much of the ESS is being found?* | *How precise is the ESS bin?* | *How much of the NESS is being found?* | *How precise is the NESS bin?* | |
| | (a*) | (b*) | (c*) | (d*) | (e*) | (f*) | (g*) | Sensitivity | Precision | Sensitivity | Precision | Accuracy |
| **Manual classification** | 48 | | | 36 | | | 16 | | | | | |
| The SEAC Method | **37** | **4** | **11** | **35** | **4** | **1** | **20** | 77.1% | 90.2% | 97.2% | 89.7% | **90.0%** |
| **Scopus** | 32 | 11 | 10 | 27 | 20 | 6 | - -* | 76.2% | 74.4% | 81.8% | 57.4% | 65.6% |
| WoS | 26 | 6 | 11 | 27 | 20 | 3 | - -* | 70.3% | 81.3% | 90.0% | 57.4% | 67.1% |
| **Sopus+WoS combined** | 43 | 16 | 5 | 27 | 14 | 9 | - -* | 89.6% | 72.9% | 75.0% | 65.9% | 70.0% |

*LEGEND

High precision > 50% = a:(a+b) or d:(d+e)

High sensitivity >80% = a:(a+c) or d:(d+f)

- -does not differentiate

**Table 7. Overall performance of the SEAC method.**

| Category | Precision | Sensitivity | Accuracy |
|---|---|---|---|
| AESS | 90% | 77% | |
| ANESS | 90% | 97% | |
| **Average** | **90%** | **87%** | **90%** |

ESS publications for the SS category in the table). The remaining rows of Table 8 show the results for five key journals in our analysis across the three citation databases. As noted above, we used three journals as training sets for our script: *Agriculture and Human Values* (as a social science journal), *Field Crops Research* (as a biophysical science journal), and *Agricultural Systems* (as a mixed journal). Two additional journals—*Agroecology and Sustainable Food Systems* and *Agriculture, Ecosystems and Environment*—are also included in our comparison in Table 8 because these two journals had the highest number of publications within our ESS and NESS corpora, respectively.

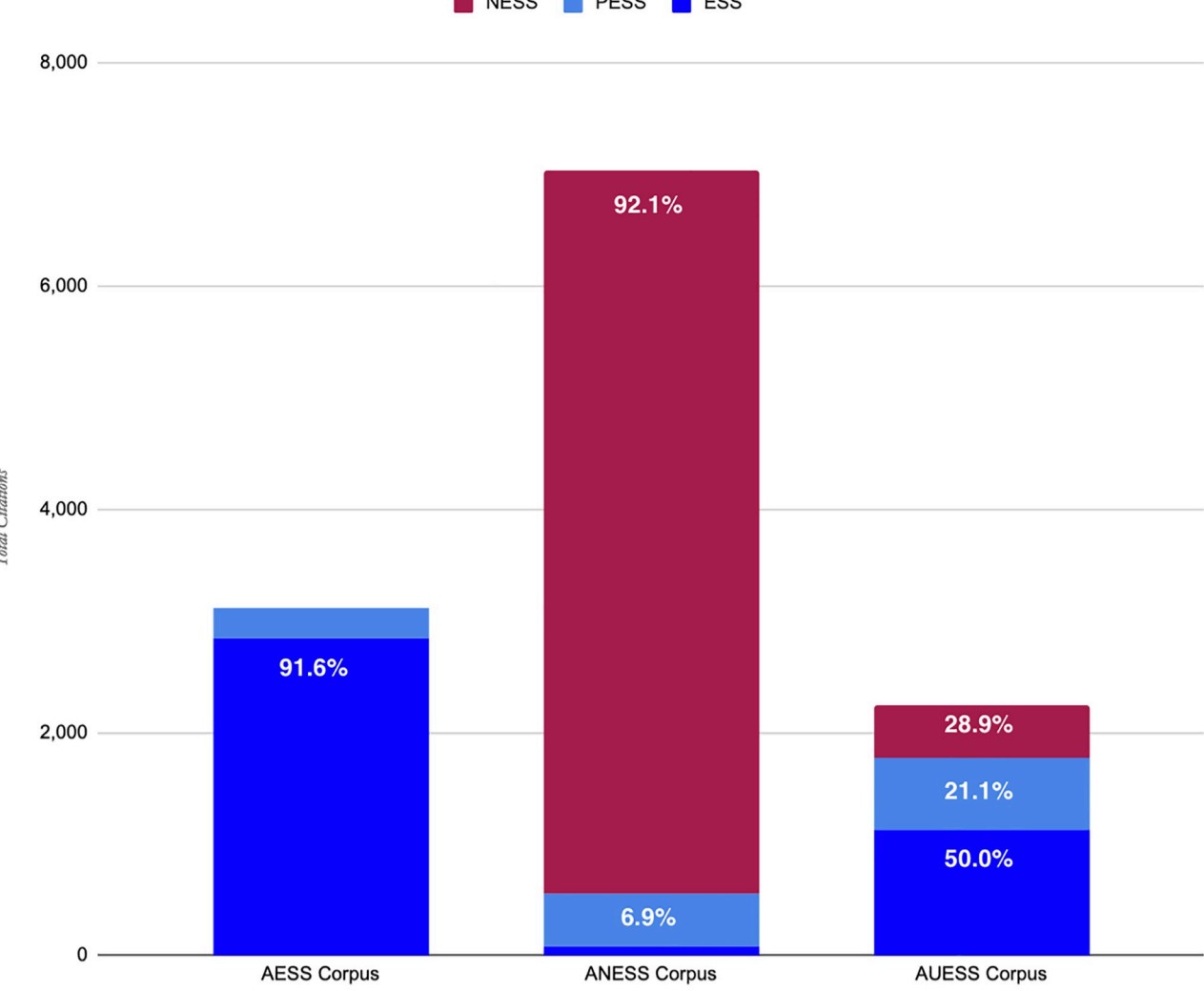

**Fig 3. Probable composition of the final three corpora (subsets of the Agroecology Corpus 2019).**

**Table 8. A comparison of agroecology literature classified as "social science" in different databases.**

| | | Scopus<br>Social Science Subject Area | WoS<br>Social Science Index (SSCI) | SEAC Method applied to Agroecology Corpus 2019 |
|---|---|---|---|---|
| **Full database** | Total # | 7,694 | 6,060 | 12,398 |
| | Total SS # | 1,405 | 1,175 | 3,125 |
| | % SS | 18% | 19% | 25% |
| *Agriculture and Human Values* | Total # | 42 | 36 | 44 |
| | Total SS # | 0 | 36 | 38 |
| | % SS | 0% | 100% | 86% |
| *Field Crops Research* | Total # | 74 | 68 | 79 |
| | Total SS # | 0 | 3 | 7 |
| | % SS | 0% | 4% | 9% |
| *Agricultural Systems* | Total # | 116 | 103 | 120 |
| | Total SS # | 0 | 37 | 56 |
| | % SS | 0% | 36% | 47% |
| *Agroecology and Sustainable Food Systems* | Total # | 135 | 138 | 142 |
| | Total SS # | 135 | 69 | 91 |
| | % SS | 100% | 50% | 64% |
| *Agriculture, Ecosystems and Environment* | Total # | 208 | 172 | 243 |
| | Total SS # | 0 | 18 | 41 |
| | % SS | 0% | 10% | 17% |

SS = "Social Science"

Table 9 shows the results of the SEAC method applied to the Scopus and WoS agroecology searches. The top rows (total and ESS) are comparable to the top rows (total and SS) of Table 8.

Table 8 shows that the SEAC method identified 25% of all agroecological literature in our database as ESS, while Scopus and WoS identified SS publications as 18% and 19%, respectively, of their databases. When we use the SEAC method on just the Scopus and WoS

**Table 9. Results of the SEAC method applied to scopus and WoS.**

| | SEAC Method applied to Scopus | SEAC Method applied to WoS | Total Unique Publications (Duplicates Removed) |
|---|---|---|---|
| Total # | 7,694 | 6,060 | 8,776 |
| Total ESS # | 2,019 | 1,739 | 2,417 |
| % ESS | 26% | 29% | 28% |
| Total NESS # | 4,258 | 3,236 | 4,727 |
| % NESS | 55% | 53% | 54% |
| Total UESS # | 1,417 | 1,085 | 1,632 |
| % UESS | 18% | 18% | 19% |

databases as seen in Table 9, it identifies 26% and 29%, respectively, of the databases as ESS. This is a 43% increase for Scopus and a 48% increase for WoS; or, put in another way, using the SEAC method, 614 more papers in Scopus were identified as ESS, as were 564 papers in WoS. Thus, when applied to specific databases, the SEAC method identifies a substantially larger body of agroecology ESS literature, which suggests a substantial under-identification of agroecology ESS literature in the major citation c databases.

Additionally, the SEAC method involves combining databases to cast a broader net than even the multidisciplinary bibliographic databases. Table 8 shows that combining databases through the SEAC method creates a considerably larger corpus than the corpora included in the main composite citation databases. Scopus identified 7,694 agroecology publications, while WoS identified 6,060. Our search identified 12,398 unique publications, which is a 60% increase from those found in the largest single database, Scopus. This suggests that when seeking to identify all publications within a transdisciplinary field like agroecology, combining the results of many databases is very important since even the multidisciplinary bibliographic databases exclude a considerable amount of peer-reviewed literature.

To further demonstrate the effectiveness of the SEAC method, we reviewed 20 random publications which produced contradicting results between multidisciplinary bibliographic databases and the SEAC method (Table 10). We selected a random sample of two types of publications: 1) publications that were classified as ESS by the SEAC method and as NESS by Scopus or WoS and 2) publications that were classified as NESS by the SEAC method and as ESS by Scopus or WoS. We manually classified those publications by reviewing the title and abstracts and found that the SEAC method was able to more accurately identify ESS publications than the multidisciplinary bibliographic databases (Table 10), even in subtle cases such as publication 5. This article, titled "*Black benniseed (Sesamum radiatum Schum. et Thonn.) cultivated as leafy vegetable in Benin*" [47], is an ethnobotanical study that could be mistaken as NESS based on the title. However, a careful reading of the abstract and keywords and review of the text shows the use of household surveys to identify cultural traditional practices in the cultivation of black benniseed; the study also mentions the role of women. Both Scopus and Web of Science classify this as NESS based on their coarser methods.

It is also useful to examine more detailed comparisons to see from where the differences arise, which is largely due to different classification systems. Scopus categorizes publications as being part of social science at the level of the journal; thus, a journal—and *all* of its contents—can either be considered social science or not social science. Thus, Scopus does not attempt to identify social science articles within largely biophysical or transdisciplinary journals. WoS differentiates social science publications (by including them in its WoS SSCI database) at the level of journals and, for some journals, individual articles. According to the description of WoS SSCI, it "fully covers over 2,900 journals across 50 social sciences disciplines. It also indexes individually selected, relevant items from over 3,500 of the world's leading scientific and technical journals" [48].

Looking at the level of individual journals in Table 8 allows us to see how undercounting in each database occurs. Scopus misses ESS publications from journals that are deemed not fully social science journals (e.g., all ESS in *Agricultural Systems* are missed since the journal as a whole is not considered a social science journal). Inversely, Scopus creates significant false positives of ESS by labeling whole interdisciplinary journals, notably *Agroecology and Sustainable Food Systems*, as ESS, even when there are clearly articles that are exclusively engaged in biophysical science. In contrast, the undercounting within WoS seems to be based on how the database identifies social science articles within multidisciplinary journals. The WoS method is considerably more conservative than the SEAC method. This under-identification of individual publications is likely somewhat offset by the same error as Scopus, i.e., the designation

**Table 10. Validating the SEAC method with a random set of disagreeing classifications.**

| | Title | Keyword(s) | Classification Method | | | |
|---|---|---|---|---|---|---|
| | | | Manual | the SEAC method | Scopus | WoS |
| 1 | *Relations Between Organic Agriculture and Agroecology: Current Challenges Around the Principles of Agroecology* | organic agriculture;agro-ecology; agroecological transition; Environmental Sciences & Ecology | ESS | ESS | - | NESS |
| 2 | *Conceptualizing a Sustainable Food System in an Automated World: Toward a "Eudaimonian" Future* | agroecology;automation; Eudaimonian food system;food security;food system;social-ecological system | ESS | ESS | NESS | - |
| 3 | *Food sovereignty and social technologies: an autonomous development experience from the Andes of Peru* | organization;profitable mountains; yachachiq;agroecology;social power;Social Sciences—Other Topics | ESS | ESS | - | NESS |
| 4 | *Viet Nam's food security: A castle of cards in the winds of climate change* | Agroecology;Climate change; Economic growth;Food insecurity; Food sovereignty;Agriculture; Chemical contamination;Climate models;Economics;Food supply; Industrial economics;Agricultural productions;Agro ecologies; Economic growths;Green economies;Industrial technology; Social and environmental; Structural constraints | ESS | ESS | NESS | - |
| 5 | *Black benniseed (Sesamum radiatum Schum. et Thonn.) cultivated as leafy vegetable in Benin* | Benin;Cultural practices;Diversity; Morphological characterisation; Sesamum radiatum;Vegetable; agroecology;crop production; cultivar;cultivation;cultural tradition;data acquisition;ethnic group;experimental study;food security;genetic analysis;genetic resource;germplasm;household survey;income;intraspecific variation;leafy vegetable; morphology;nomenclature; nutrition;participatory rural appraisal;plant breeding;species diversity;village;Benin [West Africa];Sesamum | ESS | ESS | NESS | NESS |
| 6 | *Agroecological management improves ecosystem services in almond orchards within one year* | Agroecology;Almonds;Ecosystem services;ES bundles;South-eastern Spain;Woody crops | NESS | NESS | ESS | ESS |
| 7 | *Biochar from agro-byproducts used as amendment to croplands: An option for low carbon agriculture* | Agricultural waste;Biochar; Climate change;Sequestration/release of carbon;agricultural emission;agricultural land; agroecology;alternative agriculture;carbon emission; carbon sequestration;charcoal; crop yield;emission control; emissions trading;soil amendment; yield response | UESS | NESS | ESS | - |
| 8 | *Evaluation of biochemical and yield attributes of quality protein maize (Zea mays L.) in Nigeria* | Hybrids;Inbred lines;Lysine;Open pollinated varieties;Tryptophan | NESS | NESS | ESS | N/A |

(*Continued*)

**Table 10.** (*Continued*)

| | Title | Keyword(s) | Classification Method | | | |
|---|---|---|---|---|---|---|
| | | | Manual | the SEAC method | Scopus | WoS |
| 9 | *Longitudinal analysis of maize diversity in Yucatan, Mexico: Influence of agro-ecological factors on landraces conservation and modern variety introduction* | Crop evolution;Crop genetic diversity;Milpa;Traditional varieties | NESS | NESS | NESS | ESS |
| 10 | *Diversity of alternative hosts of maize stemborers in Trans-Nzoia district of Kenya* | adaptation;agriculture;animal; article;Bacillus thuringiensis; ecology;genetics;growth, development and aging;insect control;Kenya;Lepidoptera; pathogenicity;physiology;plant; Poaceae;population dynamics; season;transgenic plant; Adaptation, Physiological; Animals;Plants, Edible;Plants, Genetically Modified;Seasons | NESS | NESS | ESS | - |

of whole journals as social science when not all publications are indeed social science (e.g., *Agriculture and Human Values*). Thus, for both Scopus and WoS, the undercounting of ESS publications is likely greater than the percentages reveal, since a considerable number of false positives offset more undercounting than initially seen in the top rows of Table 8.

## Conclusion

Our goal was to examine the body of agroecology literature to see the extent to which it engages with social sciences. Since the corpus we gathered was a three times larger than any previous review of agroecology and since pre-existing tools did not allow us to identify social science publications to our satisfaction, our first step was creating the SEAC method we have described in detail above.

In conclusion, we detail the strengths and weaknesses of the SEAC method, as well as future possible directions in its use. Its strengths include accuracy, replicability, and expandability, each detailed in subsequent paragraphs below. Following those, we detail its weaknesses in the domains of being more time- and knowledge-intensive, having a limitation that involves non-binary classification, and the need for continued maintenance. Lastly, we discuss specific future directions within which the SEAC method could be used.

The SEAC method has significantly better precision and overall accuracy, compared to the main omnibus citation databases, in determining whether a publication should be broadly classified as social science or not social science. Partly this derives from its ability to identify publications engaging in social science in journals in which they are harder to find, especially within transdisciplinary journals that are not easily categorized at the journal level as biophysical science or social science. It is also able to produce a social science classification with nearly no false positives, which will allow for more robust large-scale literature reviews and overviews.

Another strength is that the SEAC method is easily replicable with low computation power. It can be modified with minimal programmatic skills and because it uses widely available software packages, i.e., spreadsheets and Python, and it relies upon relatively simple calculations that depend on a terms list that is flexible and easily adaptable.

One of its greatest strengths is that it is easily expandable, and can be used in a variety of applications. The SEAC method is configurable and scalable to large databases, and is limited only by the computing power and capacity of the spreadsheet software used for analysis. It can be run on results from a single citation database, or any compilation thereof, as long as title, abstract, and author keywords can be downloaded as separate fields. Because it uses a non-binary classification, it can be more inclusive of transdiscipinary disciplines for which a simple social science vs. non social science distinction is not always possible. Lastly, it is expandable to other classification needs beyond social science vs. non-social science, provided that terms can be used to differentiate between publications (e.g., empirical vs. non-empirical research, field vs. laboratory research, etc).

Its first weakness that we identify is that the SEAC method is not integrated into the classification systems of any bibliographic databases, and therefore is more knowledge- and time-intensive compared to these off-the-shelf omnibus databases. The SEAC method requires prior knowledge of the discipline(s) to identify keywords that are related to that discipline, or requires consultation with experts in the discipline(s) to which it is applied. Overall, we believe the expert consultation step speeds up and improves the accuracy of the process significantly. The method also requires basic programming knowledge to run and reconfigure. These characteristics mean that it requires more time than using already existing tools. Additionally, it means that expanding the SEAC methods to other transdiscipinary fields will likely require adjusting the terms list to match the common social science terms used in those disciplines (hence the knowledge-intensive weakness above).

The SEAC method also has some technical limitations. While a non-binary classification can be helpful (hence a strength above), it also produces a relatively large corpus of unclassifiable texts, this can be reduced by changing the threshold of the total score of the Stepwise Equation. Additionally, scalability beyond 13,000 publications has not yet been tested.

For the last weakness we identify, the question of maintenance is important. As the social science engagement in agroecology evolves and produces new relevant terms, the SEAC method will likely require updates to the terms list to keep pace with the field. We would expect the same requirement for other transdiscipinary fields undergoing continuous change. However, by analyzing these changes to the list of relevant terms over time, one could gain insight into which theories, perspectives, or lines of argument were important at different points in the development of a field.

We see a number of future directions for the SEAC method. First, utilizing full-text data for each publication would improve the results, although we note that it is difficult to access without automated full-text-mining software. Others have found that full-text data showed no gains in sensitivity, very small gains in accuracy (+2%), and noticeable gains in precision (+5%) when full texts were employed [45].

Second, coupling this method with machine learning may make it more scalable and sensitive, potentially allowing the classification of publications that are partially engaged in social science and thereby creating a gradient of engagement within social science. As noted above, the non-binary aspect of the SEAC method is both a strength and a weakness, depending on one's perspective and ontological leanings. The SEAC method results in a continuous score that is non-binary. In our case, this ranged from 0 to 179. We chose cut-offs to allow us to differentiate three categories (ESS, UESS, NESS), with our main goal being the high precision of our AESS corpus, as we did not want publications that were entirely biophysically-oriented in the corpus (false positives). We left a middle ground of UESS since we wanted to acknowledge the script could not reliably categorize some publications in a binary fashion as engaged with social science or not. Even us as authors were unable to manually classify all publications in a binary fashion as a number of publications

were only marginally engaged in social science. We believe this reflects an important reality, that for some publications it is difficult to tell whether they are engaged with social science, even though this ambiguity is challenging for those who want to force a binary decision. However, since it is a continuous score, other users of the method could set different cutoffs depending on their goals. It is also possible to create other groupings based on score, such as publications with moderate scores (e.g., moderately engaged) to those of very high scores (very engaged), although we have not done so here and this would require further validation. Machine learning might make these types of classification more functional.

Third, taken together, the application of the SEAC method to the five citation databases we included shows that this method is useful in our specific application—trying to better identify agroecology's engagement with social science—and we suspect it will be useful generally in other fields at the interfaces of the biophysical and social science. To this end we have shared access to the Python script with runnable files and video tutorials (**escholarship.org/uc/item/ 5hb78383**). We are interested to know which part(s) of the script can be kept for general explorations into other transdisciplinary and interdisciplinary fields' engagement with social sciences, and which parts need to be reconfigured and/or amended for specific fields. We hope other scholars will use our open-access and open-source method to reconfigure and apply the SEAC method to other fields at the interfaces of the biophysical and social sciences, such as sustainability science and political ecology.

## Supporting information

**S1 Dataset. Agroecology Corpus 2019: The results of the Agroecology Corpus with the SEAC method applied.**
(XLSX)

## Acknowledgments

We thank members of the Galt Lab for their input and feedback at multiple points in the research process. We thank William Dowling for his support with the mathematical dimensions of the SEAC method and for improving the manuscript. We're also thankful to David Michalski and Michael Ladisch of the UC Davis Library for their help understanding various citation databases, bibliometric tools, and comparative possibilities. We thank Catherine Brinkley, Mark Cooper, and Peter Rosset for reviewing and improving this manuscript.

## Author Contributions

**Conceptualization:** Natalia Pinzón, Ryan E. Galt.

**Data curation:** Natalia Pinzón, Ryan E. Galt, Marcela Beatriz Baukloh Coronil.

**Formal analysis:** Natalia Pinzón, Ryan E. Galt.

**Funding acquisition:** Ryan E. Galt.

**Investigation:** Natalia Pinzón, Ryan E. Galt.

**Methodology:** Natalia Pinzón, Ryan E. Galt.

**Project administration:** Natalia Pinzón.

**Resources:** Natalia Pinzón, Ryan E. Galt.

**Software:** Marcela Beatriz Baukloh Coronil.

**Supervision:** Ryan E. Galt.

**Validation:** Natalia Pinzón.

**Visualization:** Natalia Pinzón, Marcela Beatriz Baukloh Coronil.

**Writing – original draft:** Natalia Pinzón, Ryan E. Galt.

**Writing – review & editing:** Natalia Pinzón, Ryan E. Galt.

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
