## [Decision Letter · Decision Letter 0]

13 Oct 2022

PONE-D-22-04014Identifying social science engagement within agroecology: Classifying transdisciplinary literature with an automated textual classification methodPLOS ONE

Dear Dr. Pinzón,

Thank you for submitting your manuscript to PLOS ONE. After careful consideration, we feel that it has merit but does not fully meet PLOS ONE’s publication criteria as it currently stands. Therefore, we invite you to submit a revised version of the manuscript that addresses the points raised during the review process.

We look forward to receiving your revised manuscript.

Kind regards,

Arkadiusz Piwowar

Academic Editor

PLOS ONE

Journal Requirements:

Reviewers' comments:

Reviewer's Responses to Questions

**Comments to the Author**

1. Is the manuscript technically sound, and do the data support the conclusions?

Reviewer #1: Yes

Reviewer #2: Yes

2. Has the statistical analysis been performed appropriately and rigorously? 

Reviewer #1: Yes

Reviewer #2: Yes

3. Have the authors made all data underlying the findings in their manuscript fully available?

Reviewer #1: Yes

Reviewer #2: Yes

4. Is the manuscript presented in an intelligible fashion and written in standard English?

Reviewer #1: Yes

Reviewer #2: Yes

5. Review Comments to the Author

Reviewer #1: I carefully read the submitted text. I want to emphasize its technical correctness and, above all, its content value.

The authors have proposed a comprehensive method of conducting bibliometric analysis with much greater sensitivity/accuracy than previous ones. Although they themselves emphasize that the SEAC method is more time-consuming and knowledge-intensive, it is characterized by significant development potential, which the authors also write about.

I hope that the SEAC method will be welcomed by the scientific community .

Reviewer #2: Please see attached file

6. PLOS authors have the option to publish the peer review history of their article (what does this mean?). If published, this will include your full peer review and any attached files.

Reviewer #1: No

Reviewer #2: **Yes: **Rachel Mason

---

## [Author Response · Author response to Decision Letter 0]

17 Nov 2022

Thank you for your thoughtful and detailed comments. We are grateful for the level of attentiveness you gave to our manuscript. We have incorporated your suggestions into a revised manuscript and have addressed your concerns in attached documents.

---

## [Decision Letter · Decision Letter 1]

29 Nov 2022

Identifying social science engagement within agroecology: Classifying transdisciplinary literature with a semi-automated textual classification method

PONE-D-22-04014R1

Dear Dr. Pinzón,

We’re pleased to inform you that your manuscript has been judged scientifically suitable for publication and will be formally accepted for publication once it meets all outstanding technical requirements.

Kind regards,

Arkadiusz Piwowar

Academic Editor

PLOS ONE

Additional Editor Comments (optional):

Reviewers' comments:

Reviewer's Responses to Questions

**Comments to the Author**

1. If the authors have adequately addressed your comments raised in a previous round of review and you feel that this manuscript is now acceptable for publication, you may indicate that here to bypass the “Comments to the Author” section, enter your conflict of interest statement in the “Confidential to Editor” section, and submit your "Accept" recommendation.

Reviewer #2: All comments have been addressed

2. Is the manuscript technically sound, and do the data support the conclusions?

Reviewer #2: Yes

3. Has the statistical analysis been performed appropriately and rigorously? 

Reviewer #2: Yes

4. Have the authors made all data underlying the findings in their manuscript fully available?

Reviewer #2: Yes

5. Is the manuscript presented in an intelligible fashion and written in standard English?

Reviewer #2: Yes

6. Review Comments to the Author

Reviewer #2: (No Response)

7. PLOS authors have the option to publish the peer review history of their article (what does this mean?). If published, this will include your full peer review and any attached files.

Reviewer #2: **Yes: **Rachel Mason

---

## [Editor Report · Acceptance letter]

25 Jan 2023

PONE-D-22-04014R1 

Identifying social science engagement within agroecology: Classifying transdisciplinary literature with a semi-automated textual classification method 

Dear Dr. Pinzón:

I'm pleased to inform you that your manuscript has been deemed suitable for publication in PLOS ONE. Congratulations! Your manuscript is now with our production department. 

Kind regards, 

on behalf of

Professor Arkadiusz Piwowar 

Academic Editor

PLOS ONE